# Photocatalytic Properties of PbMoO₄ Nanocrystals against Cationic and Anionic Dyes in Several Experimental Conditions

Francisco Nobre [1,*], Jairo Trindade [2], Marcus do Nascimento [1], Giancarlo Souza [2], Otoniel Mendes [3], Anderson Albuquerque [4,5], Júlio Sambrano [5], Paulo Couceiro [6], Walter Brito [7], Yurimiler Leyet Ruiz [8] and José Milton De Matos [2]

[1] Instituto Federal de Educação, Ciência e Tecnologia do Amazonas, Departamento de Química, Alimentos e Meio Ambiente (DQA), Campus Manaus Centro, IFAM-CMC-AM, Manaus CEP 69020-120, Brazil

[2] Laboratório Interdisciplinar de Materiais Avançados, LIMAV-UFPI, Universidade Federal do Piauí, Departamento de Química, Centro de Ciências da Natureza (CCN), Teresina CEP 69049-550, Brazil

[3] Departamento de Física, Escola Superior de Tecnologia—EST (LabFENTOM), Universidade do Estado do Amazonas, Manaus CEP 69050-020, Brazil

[4] Instituto de Química, Departamento de Química, Universidade Federal do Rio Grande do Norte (UFRN), Natal CEP 59078-970, Brazil

[5] Grupo de Modelagem e Simulação Molecular, INCTMN-UNESP, São Paulo State University, Bauru CEP 17033-360, Brazil

[6] Laboratório de Físico-Química, LFQ-UFAM, Departamento de Química, Universidade Federal do Amazonas, Manaus CEP 69077-000, Brazil

[7] Laboratório de Bioeletrônica e Eletroanalítica (LABEL), Departamento de Química, Universidade Federal do Amazonas, Manaus CEP 69077-000, Brazil

[8] Departamento de Engenharia de Materiais, Universidade Federal do Amazonas, Manaus CEP 69077-000, Brazil

* Correspondence: francisco.nobre@ifam.edu.br; Tel.: +55-92984463086

**Abstract:** This paper reports easy and fast synthesis of PbMoO₄ nanocrystals by microwave-assisted hydrothermal (MH) method at different synthesis times (1, 10, 30 and 60 min) at 100 °C. X-ray diffraction, Rietveld refinement and Raman spectroscopy confirm all characteristics of diffraction peaks and active vibrational modes of the pure scheelite structure (tetragonal, I41/a) for all synthesized PbMoO₄ nanocrystals. The optical bandgap calculated directly from the samples is close to 3.5 eV. The images collected by scanning electron microscopy show particles with mean length from 159.90(8) nm to 303.02(3) nm with greater exposure of planes (111), (100), (011) and (110). The photocatalytic activity of PbMoO₄ nanocrystals against RhB and RBBR dyes resulted in successful degradation in short time intervals using ultraviolet light, where the best performance was achieved for the PbMoO₄-10 sample, which was 29.2 and 51.8 times more effective than photolysis. The contribution of oxidant species was monitored by radical scavengers, which confirms that holes ($h^+$) are the main oxidative species in photodegradation of RhB and RBBR dyes, while reuse of the catalyst against RhB and RBBR dyes confirms high stability of the catalyst, although recycled four times.

**Keywords:** photocatalysis; lead molybdate; DFT

## 1. Introduction

Scheelite-type compounds are commonly classified as ABO₄ minerals, where A is a metal, usually a bivalent cation ($Pb^{2+}$, $Sr^{2+}$, $Ba^{2+}$ and $Ca^{2+}$) [1]. In contrast, B is an internal transition metal with valence +6 ($Cr^{6+}$, $W^{6+,}$ or $Mo^{6+}$) [2]. These compounds have been extensively investigated in recent decades for numerous purposes and applications due to their excellent catalytic [3], semiconductivity [4], photoluminescence [5], antimicrobial [6] and magnetic properties [7] and high stability.

In this context, lead molybdate (PbMoO₄) is a well-known semiconductor that exhibits a tetragonal structure (I41/a) with four formulas per unit cell (Z = 4) [8]. The crystalline

structure for the tetragonal unit cell of $PbMoO_4$ is composed of $[PbO_8]$ clusters with deltahedral symmetry, where the lead atoms (Pb) are coordinated to eight oxygen atoms, resulting in irregular polyhedral [8,9]. Moreover, the molybdenum atoms (Mo) coordinate with four oxygen atoms, resulting in formation of clusters with tetrahedral arrangement $[MoO_4]$ of *Td* symmetry [10].

There are several methodologies to obtain lead molybdates, generally involving reactions in a solid state under high temperatures, as well as in an aqueous solution by electrostatic attraction of precursor ions, followed by formation of insoluble precipitate [11]. Therefore, the reactions commonly used to obtain $PbMoO_4$ nanocrystals are microwave-hydrothermal—MH [12], microwave-solvothermal—MS [13], conventional hydrothermal (HC), sonochemistry—SC [14] and co-precipitation method—CP [15].

When processed by different synthetic methods, the nucleation and growth process in the crystallization stage of lead molybdates results in different properties [15–17]. Thus, the surface energies ($E_{surf}$) [18], presence of crystalline defects [19], oxygen vacancies ($V_o^z = V_0^x$, $\dot{V}_0$ or $\ddot{V}_0$), degree of order/disorder of the clusters, morphology and particle size are correlated with this process [20].

Gurgel et al. [14] report $PbMoO_4$ nanocrystals by co-precipitation and sonochemistry methods, evaluating the correlation between the synthesis method and the optical and morphological properties exhibited by the obtained materials. In addition, a decrease in particle size was confirmed with addition of time in sonochemical synthesis and strong photoluminescent emission in the green region.

Du et al. [21] obtained $PbMoO_4$ nanoparticles using conventional hydrothermal synthesis at 200 °C under 12 h and investigated their catalytic properties. These authors reported high efficiency in photocatalysis of aqueous solutions of methylene blue (MB) dye (85.3%) under visible light. At the same time, Bomio et al. [9] correlated the surface energy of the planes in $PbMoO_4$ nanocrystals obtained by chemical precipitation with catalytic efficiency in degradation of RhB dye molecules in an aqueous solution. Therefore, they confirmed dependence of catalytic efficiency with surface energy related to exposure of surface planes (111), (100), (011) and (110).

Regarding stability of lead molybdates over the photo-oxidative process in an aqueous solution, Dai et al. [18] reported synthesis of pure lead molybdates and the lead molybdates fullerene supported applied in photocatalysis of rhodamine B dye (RhB). After five consecutive circles, we confirmed that the photocatalytic performance decreased by about 8%, where only 0.0049% of lead ions were detected in the solution using graphite furnace atomic absorption spectrometry. Similarly, Hashim et al. [17] reported high photostability of dendrites-like $PbMoO_4$ with exposed (001) facet after four cycles in photodegradation of RhB dye under sunlight irradiation.

This paper reports an easy and fast way to synthesize $PbMoO_4$ nanocrystals. The parameters used for the microwave-hydrothermal method were times of 1, 10, 30 and 60 min and a constant temperature of 100 °C in the absence of surfactant compounds. Thus, the experimental and theoretical study of the synthesis was correlated with the sructural, morphology, optical and catalytic characteristics in degradation of cationic compounds (RhB) and anionic (RBBR) in the presence of UV irradiation.

## 2. Materials and Methods

### 2.1. Synthesis of PbMoO₄ Nanocrystals

We synthesized $PbMoO_4$ nanocrystals using the MH method following the typical steps: 1 mmol of lead nitrate—$Pb(NO_3)_2$ (Sigma-Aldrich, São Paulo/SP, Brazil, purity > 99.0%) and 1 mmol of sodium molybdate dihydrate—$Na_2MoO_4 \cdot 2H_2O$ (Sigma-Aldrich, Sigma-Aldrich, São Paulo/SP, purity > 99.0%) were solubilized separately in Falcon tubes (15 mL capacity) with 15 mL of distilled water. Then, after solubilization of the salts, these solutions were mixed in a Teflon cup (80 mL capacity), adding 20 mL of distilled water [22,23]. After, we adjusted the pH of the solution to 11, adding 1 mol $L^{-1}$ of NaOH solution, and submitted it to microwave-assisted synthesis using the Panasonic microwave

adapted system (2.45 GHz and maximum output power of 800 W), as mentioned by Silva et al. [24], at $100 \pm 3\,°C$ at different times (1, 10, 30 and 60 min), which is denoted $PbMoO_4$-1, $PbMoO_4$-10, $PbMoO_4$-30 and $PbMoO_4$-60, respectively. In this procedure, the pressure of system is closed to $2.7 \pm 0.1$ atm. After cooling, the precipitate of each synthesis was collected by centrifugation (10,000 rpm for 3 min) and washed several times with distilled water. The white precipitate obtained was dried for 24 h at $80\,°C$.

### 2.2. Analytical Methods

#### 2.2.1. XRD Measurement and Rietveld Refinement of $PbMoO_4$ Nanocrystals

The structural characterization by X-ray diffraction analysis (XRD) was performed using a Shimadzu diffractometer, XRD 6000 model with CuK$\alpha$ ($\lambda$ = 0.15406 nm), collecting diffraction data in the 2$\theta$ from 5° to 80° with a step size of 0.02°. In this study, we performed the structural Rietveld refinement method using the Fullprof package software [19], July 2022 version. We refined the lattices parameters, atomic coordinates, background, occupational factor, isotropic thermal factor and the Caglioti function ($U$, $V$ and $W$), among other parameters. All diffraction peaks of the samples were adjusted using the Pseudo–Voigt—PV function [6].

#### 2.2.2. Vibrational Raman Spectroscopy

The active vibrational modes were studied using Raman spectroscopy, collecting the spectra of the samples operating a confocal Bruker Raman microscope, SENTERRA model, São Paulo, Brasil, coupled with a Charge Coupled Device (CCD) module to cool the system. All spectra were collected from 50 to 1600 cm$^{-1}$ using a green laser ($\lambda = 523$ nm) for sample excitation and output power of 0.5 mW.

#### 2.2.3. UV–Vis by Diffuse Spectroscopy (DRS)

The UV–vis spectra of the solid samples were collected using a Shimadzu spectrophotometer, UV2600 model, São Paulo, Brasil, coupled with diffuse reflectance modulus. In general, we recorded the DRS spectrum of each sample from 200 to 900 nm, with a scan speed of 10 nm s$^{-1}$. Barium sulfate—$BaSO_4$ (Sigma-Aldrich, >99.99%) was used as the analytical standard of reflectance.

#### 2.2.4. Field Emission Scanning Electron Microscopy—FE-SEM

The FE-SEM images were used to study the synthesized nanoparticles' morphology and distribution of size length. These were collected by field-emission scanning electron microscope (FE-SEM) from FEI Company, Quanta FEG 250 model, São Paulo, Brasil.

#### 2.2.5. Photocatalytic Performance of $PbMoO_4$ Nanocrystals under Degradation of Cationic and Anionic Dye

We carried out the photocatalytic performance of $PbMoO_4$ nanocrystals in a typical procedure using 50 mL of the rhodamine B dye solutions (RhB, = 4.7 ppm) or Remazol Brilliant Blue R (RBBR, = 200 ppm) with 50 mg of the $PbMoO_4$ nanocrystals as photocatalyst irradiated with three UV lamps ($\lambda$ = 253.7 nm) with an output power of 45 W each ($3 \times 36\,W = 108\,W$). In the kinetic study, the maximum wavelength for RhB dye ($\lambda$ = 554 nm) and RBBR ($\lambda$ = 665 nm) was collected, measuring the aliquots (volume = 0.5 mL) under consecutive intervals along 90 min for RhB and 120 min for RBBR dye. We centrifuged the suspension and then measured the wavelength of the dye solutions using a Thermo Scientific Spectrophotometer, Genesys 10S model, São Paulo, Brasil, at a scanning speed of 10 nm s$^{-1}$ using quartz cuvettes to accommodate the solutions.

### 2.3. Computational Methods for $PbMoO_4$ Nanocrystals

The lead molybdate ($PbMoO_4$) was modeled in a bulk phase from its most stable structure at ambient conditions, the tetragonal scheelite-type structure [20] (space group 88, $I41/a$), considering the internal and translational symmetry of the crystal through

periodic 3D conditions. In this structure each $Mo^{6+}$ cation is coordinated to four $O^{2-}$ anions, and each $Pb^{2+}$ cation is coordinated to eight oxygens being formally described as $[MoO_4]$ and $[PbO_8]$ clusters. The lattice parameters ($a = 5.431(2)$ Å, $c = 12.106(5)$ Å and $V = 357.1(2)$ Å$^3$) and atomic positions from reference single crystal [25] of tetragonal $PbMoO_4$ unit cell were used as initial guesses.

We investigated the structural, vibrational and electronic properties through periodic first-principles calculations in the framework of density functional theory (DFT) using the CRYSTAL 17 package [26]. The B3LYP hybrid functional [27,28], including dispersion corrections through Grimme D3 potential [29], was used for all calculations. Hybrids functionals, such as B3LYP, have been successfully employed to describe many properties in several strongly correlated systems because they partially eliminate the self-iteration error inherent to the standard DFT formalism, besides serving as a reference to computational calculations of many oxides in periodic defective systems [30,31], including $PbMoO_4$ [32,33]. Allll-electron triple-zeta 8/411/1 (s/sp/d) basis set was used for oxygens, and Hay & Wadt large core (HAYWLC) 211/3 (sp/d) and small core (HAYWSC) 311/31 (sp/d) pseudopotentials were used for Pb and Mo, respectively, as available in the Crystal Basis Set Library [33].

In order to understand the electronic structure of the reduced system, we removed one oxygen from a fully relaxed $2 \times 2 \times 1$ supercell ($a = 10.8(3)$ Å, $c = 11.8(2)$ Å and $V = 1386.7(7)$ Å$^3$), containing 16 units of ($PbMoO_4$), forming a system with 1.56% of oxygen vacancy ($PbMoO_{3.9375}$). The stoichiometric (super) cells are closed-shell systems and were treated under restricted Kohn–Sham (KS) formalism. In contrast, we treated the reduced cells under the unrestricted KS formalism to better describe the electrons left by the removed oxygen. The spin multiplicity (singlet S0 and triplet T1) was also checked for stoichiometric and reduced systems.

The accuracy for the Coulomb and exchange series was controlled by five tight thresholds parameters set to (ITOL$_{1-4}$ = $10^{-10}$; ITOL$_5$5 = $10^{-20}$). The shrinking factor (Pack–Monkhorst and Gilat net) was set to 6 (or 2) for unit cell (or supercell), corresponding to 36 (or 8) independent k-points in the fundamental part of the Brillouin zone integration in the primitive cell.

The SCF criteria convergence was governed by a threshold on the energy of $10^{-10}$ and $10^{-11}$ Hartree for geometry optimizations and vibrational calculations, respectively. The stoichiometric structures were fully optimized (atomic positions and lattice parameters) with a quasi-Newtonian technique combined with the Broyden–Fletcher–Goldfarb–Shanno (BFGS) algorithm for Hessian updating, taking very tight criteria for convergence on gradient (0.0001 a.u.) and nuclear displacements (0.0004 a.u.). In contrast, for the reduced supercell, the lattice parameters were fixed, and the atomic positions were relaxed. This methodology is acceptable for low concentrations of point defects.

Infrared and Raman normal modes and their corresponding harmonic frequencies were obtained from tightly optimized stoichiometric bulk. The frequencies were computed at the $\Gamma$ point by diagonalizing the mass-weighted Hessian matrix. We analytically computed the Raman spectra with relative intensities of the peaks by exploiting a scheme based on the solution of first and second-order Coupled-Perturbed-Hartree–Fock/Kohn-Sham (CPHF/KS) equations [34], and we obtained the simulated spectrum of powder samples taking into account all possible orientations of crystallites [35], providing a fingerprint by which the contribution of each cluster vibration can be detailed described. The modes were visualized through the J-ICE program [36].

We studied the electronic structure through atomic charges (Mulliken population analysis and Hirschfeld-I partitioning scheme), band structure and density of states (DOS) using the Properties 17 module of the CRYSTAL code considering the same k-point sampling as that used during the diagonalization of the Fock matrix. Electronic band structure was obtained at the appropriate high-symmetry path in the first Brillouin zone. The density of states (DOS) was calculated using the Fourier−Legendre technique with a polynomial degree of 12.

The graphical manipulations were performed using molecular graphics programs XCrySDen [36] and Vesta [37].

## 3. Results and Discussion

### 3.1. XRD Pattern and Rietveld Refinement

Figure 1a shows the experimental diffraction patterns of $PbMoO_4$ nanocrystals synthesized by the MH method for 100 °C at 1, 10, 30 and 60 min.

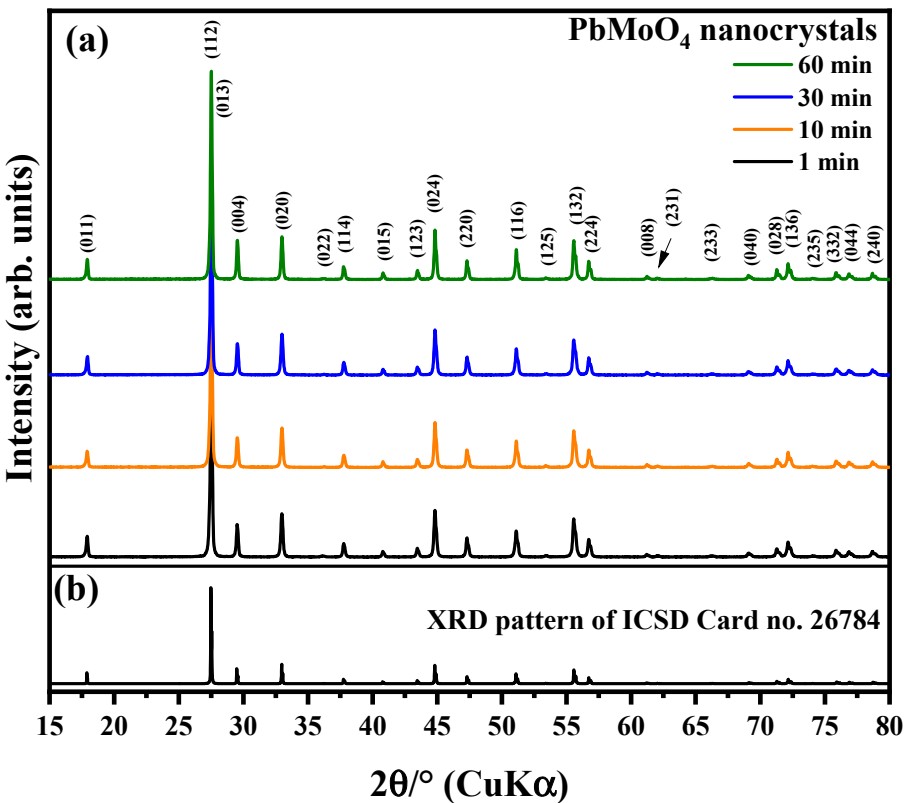

**Figure 1.** XRD pattern of (**a**) $PbMoO_4$ obtained from MH synthesis and (**b**) ICSD card no. 26784. The vertical lines indicate the Bragg peak position.

All XRD peaks found in the diffraction pattern of synthesized samples agree with the Inorganic Crystal Structure Database (ICSD) card no. 26784 [25] literature [6,38]. Based on this indexation, we confirmed that these samples have a pure tetragonal structure with Scheelite-type arrangement, space group of I41/a, point-group symmetry of $C_{4h}^6$ and four formulas per unit cell, Z = 4 [39].

The $PbMoO_4$ samples have well-defined peaks, indicating that the formation process has resulted in high crystallinity degree at short and long range [40]. In addition, we have not found XRD peaks of secondary phases or precursors used in the synthesis. That confirms the ideal chemical stoichiometry proposed for all processed reaction conditions adopted [40,41].

Figure 1b shows the XRD pattern of ICSD card no. 26784 plotted for comparison. This pattern was created using the crystallographic data as an input file for the Visualization Electronic Structure and Structural Analysis—VESTA software, version 3 [42]. It pointed out that the graphical profile of the crystallographic data contained in the card agrees with the graphical profile for the diffraction patterns of the $PbMoO_4$ nanocrystals synthesized in the present study.

The diffraction patterns obtained for the $PbMoO_4$ were used as input crystallographic information for structural refinement by the Rietveld method [43]. This methodology consists of the computed intensities of the experimental ($Y_{obs}$) and theoretical ($Y_{calc}$) diffraction data; in the present study, we used ICSD card no. 26784 as the theoretical data.

We adjusted the background of the diffraction pattern of all samples using a polynomial with six coefficients, that is, degree six. The quality of the structural refinement was evaluated by the convergence of the refined parameters checked by R_parameters ($R_p$, $R_{wp}$ $R_e$, $S$ e $\chi^2$) [42,44].

Figure 2a–d shows the Rietveld refinement plot for the PbMoO$_4$ samples synthesized in this study. These results confirmed a pure tetragonal phase for all PbMoO$_4$ samples, similar to those reported by the literature [13,17,45]. Moreover, the profile of the residual line ($Y_{obs} - Y_{calc}$) shows minimum residue, suggesting a good correlation between the experimental and theoretical data [14].

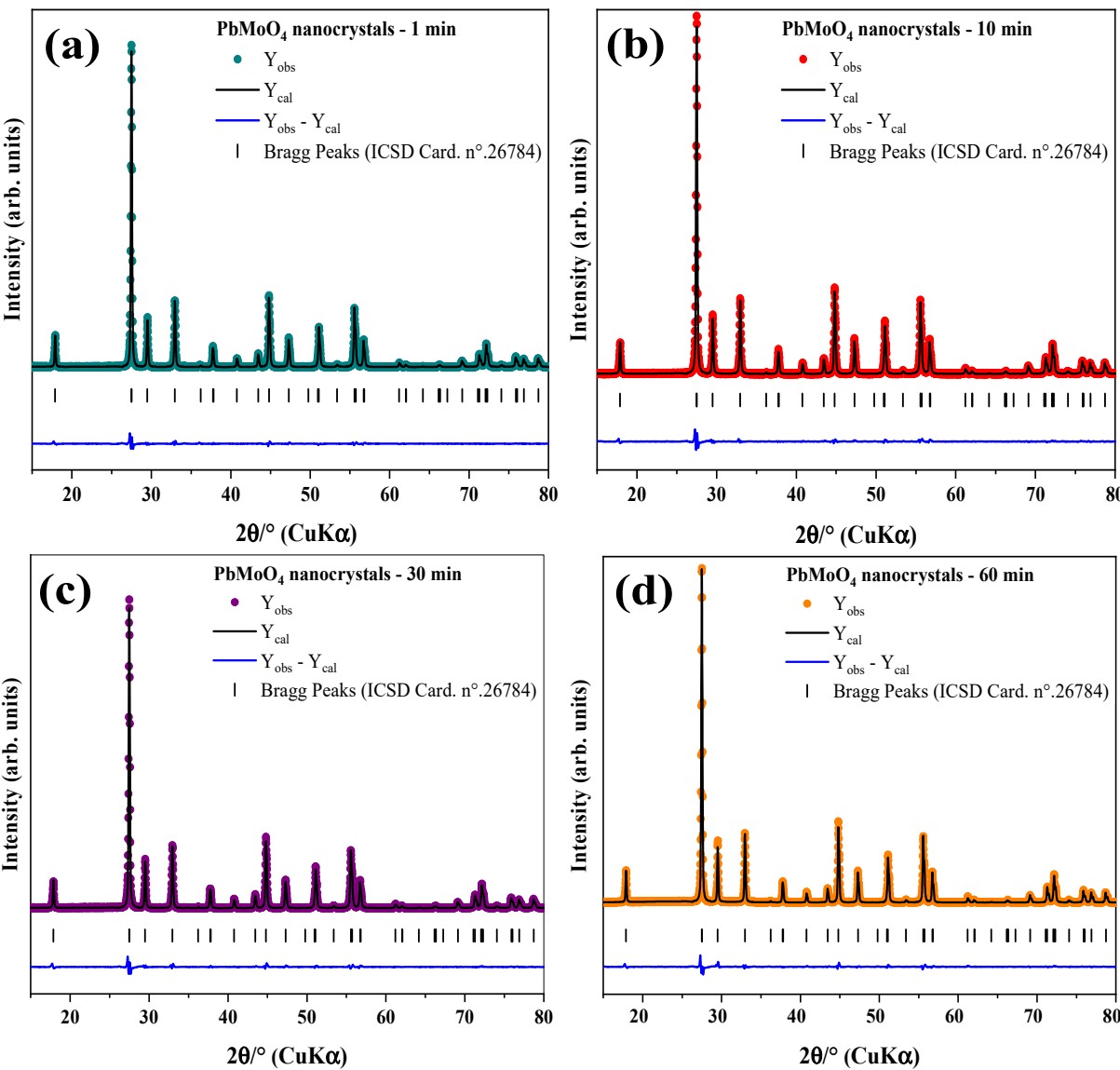

**Figure 2.** Rietveld refinement of the PbMoO$_4$ nanocrystals obtained from the MH method: (**a**) PbMoO$_4$-1, (**b**) PbMoO$_4$-10, (**c**) PbMoO$_4$-30 and (**d**) PbMoO$_4$-60.

Table S1 (available in the Supplementary Materials) summarizes the complementary results obtained from the structural Rietveld refinement regarding the atomic position, occupation and anisotropic thermal factor. Note that there was significant variation in atomic positions, mainly for the oxygen atoms, caused by the mechanism of nucleation and growth of nanoparticles [8,9,14]. The increase in synthesis time leads to different occupancy levels and anisotropic thermal factor values for the present atoms (Pb, Mo and

O). However, those agree with the values reported by the consulted literature [8] and card ICSD no. 26784.

We investigated the effect of synthesis time on lattice parameters ($a = b$ and $c$) and unit cell volume (V), as shown in Table 1. Based on these results, the crystallographic dimensions of the unit cell for $PbMoO_4$ result in a constant length of axes a and b ($a = b = 5.433(2)$ Å). On the other hand, there is a slight decrease in coordinate axis $c$, resulting in 12.103(9) Å ($PbMoO_4$-1), 12.103(7) Å ($PbMoO_4$-10), 12.102(7) Å ($PbMoO_4$-30) and stabilization at 12.1027 Å ($PbMoO_4$-60).

**Table 1.** Method, synthesis time and temperature, lattices parameters and unit cell volume (V) of the $PbMoO_4$ nanocrystals synthesized and reported by the references (Ref).

| Method | Time (min) | Temp. (°C) | Lattices Parameters Å | | V (Å$^3$) | Ref |
|---|---|---|---|---|---|---|
| | | | *a=b* | *c* | | |
| MH | 1 | 25 | 5.433(2) | 12.103(9) | 357.30 | This work |
| MH | 10 | 25 | 5.433(2) | 12.103(7) | 357.30 | This work |
| MH | 30 | 25 | 5.433(3) | 12.102(7) | 357.29 | This work |
| MH | 60 | 25 | 5.433(3) | 12.102(7) | 357.28 | This work |
| HC | 120 | 150 | 5.4350 | 12.1066 | 357.63 | [6] |
| HC | 1200 | 160 | 5.4336 | 12.1104 | 357.456 | [16] |
| SR | 720 | 1150 | 5.4351 | 12.1056 | 357.60 | [38] |
| ICSD n°. 26784 | - | | 5.4312 | 12.1065 | 357.12(24) | - |

Legend: SC = sonochemistry route; HC = hydrothermal synthesis; SR = solid-state reaction.

The average crystallite size was calculated using the Debye–Scherrer equation as presented in Equation (1) [46], where $k$ is the constant associated with the structure factor, $k = 0.9$ (spherical) and $\lambda$ is the wavelength of the copper irradiation ($CuK\alpha = 0.15406$ nm) used to collected the diffraction pattern of the samples [13,47].

$$\overline{D}_{hkl} = \frac{k\lambda}{\beta \cos\theta} \tag{1}$$

The full width at half maximum (FWHM) diffraction peaks ($\beta$) was corrected using Equation (2) for each angle of diffraction ($\theta$), where $\beta_{exp}$ and $\beta_{inst}$ are the FWHM for the experimental diffraction peaks obtained for each sample and instrumental, respectively. The instrumental FWHM was obtained from the Rietveld refinement performed for pure silicon diffraction pattern [48].

$$\beta = \sqrt{\beta_{exp}^2 - \beta_{inst}^2} \tag{2}$$

Figure 3 shows the plot of crystallite size against unit cell volume for (a) $PbMoO_4$-1, (b) $PbMoO_4$-10, (c), $PbMoO_4$-30 and (d) $PbMoO_4$-60.

The average crystallite sizes ($D_{hkl}$) calculated for $PbMoO_4$-1, $PbMoO_4$-10, $PbMoO_4$-30 and $PbMoO_4$-60 samples were 58, 54, 56 and 60 nm, respectively (see Figure 3). The variations observed for the mean crystallite size are related to nanocrystals' nucleation, formation and growth mechanism, generally described by the Ostwald ripening model [49]. The decrease in the average size from 1 to 10 min of synthesis is associated with redissolution of the nanoparticles, while an increase in the mean size of the nanoparticles occurred from 10 to 60 min due to the agglomeration of nanocrystals [3]. It can be associated to the heating process and pressure attributed in the microwave-hydrothermal system at different synthesis times.

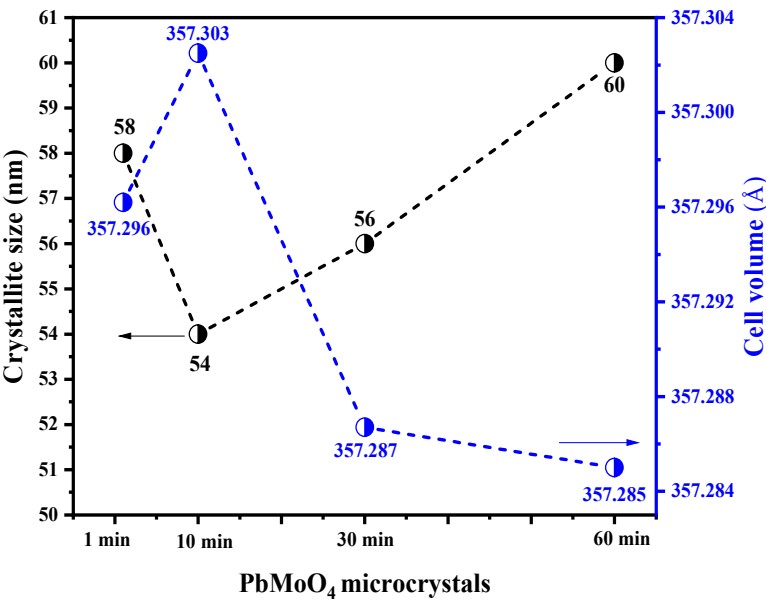

**Figure 3.** Crystallite size and unit cell volume of the PbMoO$_4$ nanocrystals obtained from the MH method for PbMoO$_4$-1(1 min), PbMoO$_4$-10 (10 min), PbMoO$_4$-30 (30 min) and PbMoO$_4$-60 (60 min).

### 3.2. Micro-Raman Vibrational Spectroscopy

Figure 4a shows the experimental Raman spectra of the synthesized PbMoO$_4$, while Figure 4b,c shows the Raman spectra obtained by DFT theoretical study in the spectral range of 50 to 1000 cm$^{-1}$.

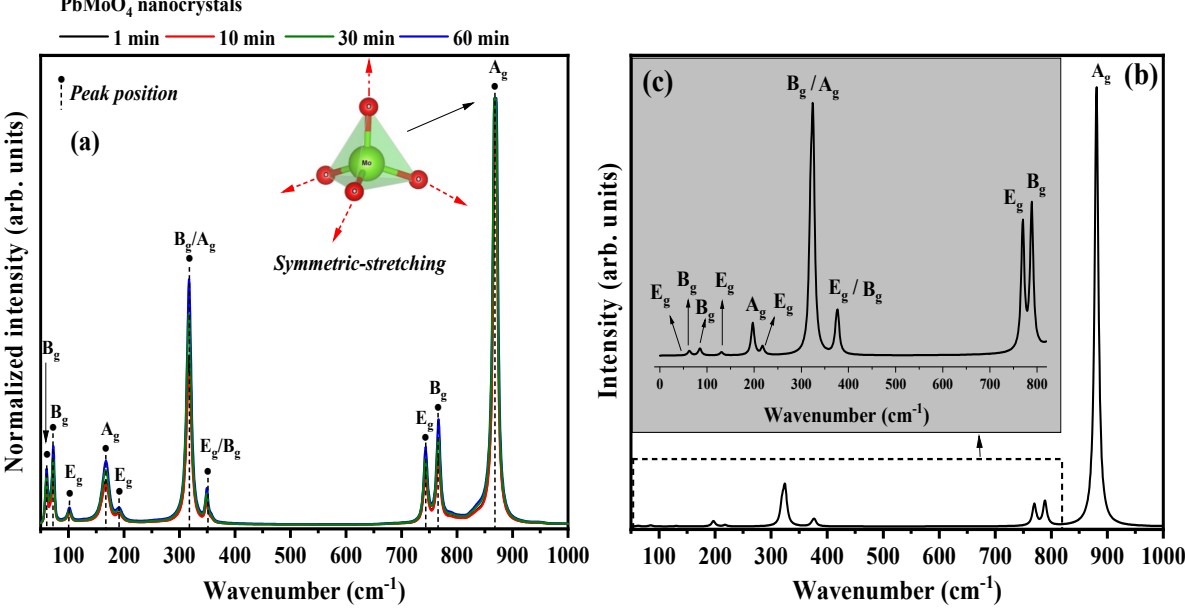

**Figure 4.** Raman spectra of the PbMoO$_4$ nanocrystals prepared by (**a**) MH method at different times (1, 10, 30 and 60 min) at 100 °C. (**b**,**c**) Raman spectra obtained from the theoretical study by DFT (B3LYP). The vertical lines indicate the peak positions of the Raman active modes.

Scheelite-type molybdates are generally reported to have two main types of molecular vibrations, internal and external modes [9]. The external modes are related to phonons coming from the crystal lattice by molecular vibrations of the bonds (O-Pb-O) of deltahedral [PbO$_8$] clusters that exhibit symmetry $D_{2h}$ [6,8,14]. Differently, the internal modes are associated with the bonds (O-Mo-O) contained in tetrahedral clusters [MoO$_4$] of symmetry $T_d$ [9,13,14,45].

Bomio et al. [9] described that the tetrahedral clusters [MoO$_4$] with the $T_d$ group have cubic symmetry, composed of four internal active modes, these being $v_1$ ($A_1$), $v_2$ ($E_1$), $v_3$ ($F_1$) and $v_4$ ($F_2$), where $F_1$ are the rotational and $F_2$ translational modes. Although there is no direct relation between the point group of symmetry $T_d$ for the molecule with the point group $C_{4h}^6$ of the unit cell For the Scheelite minerals, there is a reduction of these by point group $S_4$, the latter the subgroup of $T_d$ and $C_{4h}^6$.

In the experimental spectra for the synthesized PbMoO$_4$ nanocrystals, we identified 10 of the 13 active modes characteristic of the tetragonal structure (scheelite) for PbMoO$_4$. The active mode ($A_g$) associated with the symmetrical stretches of the bonds (O←Mo←O)/(O→Mo→O) at 867 cm$^{-1}$, asymmetric stretching (O←Mo←O)/(O←Mo←O) in the 744 ($E_g$) and 765 cm$^{-1}$ are attributed to the tetrahedral clusters [MoO$_4$] [45].

The two active modes identified from 300 to 400 cm$^{-1}$, specifically at 315 ($B_g$) and 352 cm$^{-1}$ ($B_g$), are associated with the molecular vibrations of the MoO$_4^{2-}$ in regular tetrahedra [9]. In addition, the two peaks identified in 165 ($A_g$) and 192 cm$^{-1}$ ($E_g$) are reported by the literature as being pertinent to the rotational modes ($F_1$) [6]. Finally, the modes identified in the 60 ($B_g$), 71 ($B_g$) and 101 cm$^{-1}$ ($A_g$), are all reported by the literature as pertinent to the translational modes ($F_2$) characteristic of the tetragonal structure of lead molybdate [6,8,9,14].

Table S2 (please see Supplementary Materials) summarizes the wavenumbers found in experimental Raman spectra for the PbMoO$_4$ nanocrystals synthesized and reported by the literature [6,8]. In summary, there were slight differences associated with deformations in the bonds and angles of [PbO$_8$] and [MoO$_4$], clusters, oxygen vacancies ($V_O^x$) and order/disorder of structures.

### 3.3. UV–Vis Spectroscopy by Diffuse Reflectance

Optical characterization by UV–vis using diffuse reflectance has been applied in order to evaluate the optical bandgap ($E_{gap}$) of solid-state semiconductors [45,47].

Bomio et al. [9] report the synthesis and theoretical study using the density functional theory (DFT) adopting the density of occupied states (DOS), confirming that the direct allowed transitions type ($n = 0.5$) prevailing in PbMoO$_4$ oxides. In addition, the O2$p$ orbitals strongly contribute to the density of electrons present in the valence band, as well as the Mo4$d$ and Pb6$d$ orbitals, with the conduction band.

Therefore, we estimated the $E_{gap}$ values for the PbMoO$_4$ nanocrystals synthesized using the model described by Kubelka–Munk [50,51], as shown in Equation (3), considering direct allowed electronic transitions ($n = 0.5$) [24,52–54].

$$[F(R_\infty)h\nu]^2 = C_2\big(h\nu - E_{gap}\big) \tag{3}$$

When $h\nu$ is equivalent to photon energy, $C_2$ is a proportionality constant and $F(R_\infty)$ is the Kubelka–Munk function, where $R_\infty$ is the absolute reflectance obtained by the relation $R_\infty = R_{sample}/R_{standard}$), with $R_{sample}$ as the reflectance percentage of each sample and $R_{standard}$ as the percentage reflectance of the standard [55]; the present study used barium sulfate—BaSO$_4$ (Sigma-Aldrich, purity $\geq$ 99.99%) as the standard reflectance.

The $E_{gap}$ values were estimated by plotting $[F(R_\infty)h\nu]^2$ against $E_{photon}$, then extrapolating the straight section of the paraboloid curve, obtaining the $E_{gap}$ ($x-$axis) when adopting $y = 0$ [56,57]. Figure S1a–d (please see Supplementary Materials) shows the graphs and respective $E_{gap}$ values for each of the samples composed of synthesized PbMoO$_4$ nanocrystals.

Therefore, from the results shown in parts of Figure S1a–d, we did not observe significant variations in the values of $E_{gap}$ with an increase in synthesis time to 3.2(5), 3.2(4), 3.2(5) and 3.2(4) eV referring to the PbMoO$_4$ nanocrystals synthesized at 1, 10, 30 and 60 min, respectively.

All values obtained experimentally in the present study agree with those reported by the literature consulted [1], although obtained by other synthesis routes. In particular,

Bi et al. [58] obtained $E_{gap} = 3.3$ eV for $PbMoO_4$ synthesized through the solvothermal method at 160 °C for 12 h. Gurgel et al. [14] reported a study in obtaining nanocrystals and nanocrystals of $PbMoO_4$ using the co-precipitation method, resulting in the $E_{gap}$ value of 3.23 eV.

*3.4. Atomic Structure and Electronic States*

The atomic positions and lattice parameters of lead molybdate were optimized at the B3LYP-D3 level under different situations. The calculated values for the ground state ($a = 5.41(68)$ Å, $c = 11.82(01)$ Å and $V = 346.8(2)$ Å$^3$) differ less than 3% from the single crystal reference [29], conforming to the experimental crystal in thermodynamic equilibrium. The associated electronic band gap of the unit cell was 3.54 eV (~X→P, indirect band gap), in close agreement with the experimental value of 3.2(4) eV. The differences between experimental and absolute theoretical band gaps are expected because of the semi-empirical nature of DFT functionals and the limitations of finite basis sets. We should observe the differences between the systems treated here at the same level of theory and the trends caused by structural modifications, like oxygen vacancy creation (Figure S2).

Table S3 (please see Supplementary Materials) shows the lattice parameters, the energy difference and the electronic band gap of stoichiometric and reduced $PbMoO_{4-x}$ under many spin polarization situations. For the stoichiometric $PbMoO_4$, the ground state corresponds to a singlet ($S_0$) solution (Figure S3a). The atomic charges from Mulliken population analysis and Hirshfeld-I partitioning scheme were also calculated for stoichiometric and defective bulk. Both calculated charges described the same trends, yet far from the formal charge values for Mo and O, as expected for oxides with a certain covalence degree. The net charges from Hirschfeld-I are $q = 1.75$ for Pb (close to the classical expected value, $Pb^{2+}$), $q = 2.33$ for Mo and $q = -1.02$ for O. The unexpected charges for Mo and O are explained by the $[MoO_4]$ covalence.

The triplet ($T_1$) spin-polarized solutions presented higher energy than $S_0$, and the two main possibilities are: (i) in the first one, all structural parameters from $S_0$ were maintained (Figure S3b); and (ii) in the second case, the structure was fully relaxed and the $[MoO_4]$ symmetry was broken: $[MoO_4]_o \rightarrow [MoO_4]_d$. With the tetrahedron distortion $[MoO_4]_o \rightarrow [MoO_4]_d$, the spin density ($\mu$) of Mo at $T_1$ state change from $\mu = 0.421$ to 0.863. For the Pb on $[PbO_8]$, the spin density changed from $\mu = 0.221$ in $[PbO_8]_o$ to 0.390 in $[PbO_8]_d$.

Figure S2 shows the primary energy differences between these systems, where it is possible to see clearly that a local disorder over the Mo polyhedron is necessary to achieve the triplet state. We noted that the difference between fully relaxed $S_0$ and $T_1$ structures is only ~60 meV, while for the vertical transition between a frozen unit cell, that difference is ~330 meV, indicating that disordered systems can accommodate this kind of electronic configuration better than ordered ones. When compared with ground $S_0$ unit cell, the fully relaxed $T_1$ presents a significant cell contraction. Under experimental conditions, the real crystal is not expected to present this excitation over all the extensions but only on crystallite edges.

When the oxygen vacancy ($V_O$) is formed in $[MoO_4]$, the long-range order is maintained and only the first neighbor's clusters are distorted as a result of decreased coordination other than the polaronic distortion due to cation reduction (Figure S2). The polaronic distortion was calculated in two ways: (i) considering the only atomic position relaxation, keeping the atomic lattice parameters frozen (estimated in ~20 meV) and (ii) allowing a complete relaxation, which includes the cell expansion (estimated in ~20 meV too). The polaronic distortion originated in $PbMoO_{4-x}$ is the reduction of one $Mo^{6+}$ around $V_O$ to one $Mo^{4+}$ (Figure S3a), where the spin density of reduced Mo is $\mu = 1.771$, which implies that both electrons from $O_0^x \rightarrow V_0^{\bullet\bullet} + 2e$ are trapped by only one Mo: $Mo^{6+} + 2e \rightarrow Mo^{4+}$. Another possibility, not found, is the reduction of $2Mo^{6+} + 2e \rightarrow 2Mo^{5+}$ with the formation of $V_O$, instead of $Mo^{6+} + 2e \rightarrow Mo^{4+}$. The last solution was recently found to reduce $MoO_{3-x}$ [59]. Another effect of VO is a slight cell expansion, mainly in [100] direction.

The trapped electrons in $[MoO_4]_d$ (shown through isosurfaces in Figure S3(a,b) are also responsible for the midgap states in the electronic band structures (Figure 5). The Figure 5(a–d) is projected the density of states, which that the top of the valence band (VB) of stoichiometric $PbMoO_4$ ($S_0$) is composed mainly of O 2p (red) contribution. In contrast, the conduction band bottom (CB) is composed mainly of Mo4$d$ (green), with a low contribution of Pb6$p$ (yellow). The trapped spin on symmetry brooked $PbMoO_4$ ($T_1$) occupy a midgap state located in $[MoO_4]_d$, 1.89 eV below the CB, specifically in Mo4$d_{z^2}$, generating a conductor (metallic behavior). The band gap decrease with polyhedral distortion helps to explain the modifications on the experimental band gap and photoluminescence of disordered materials.

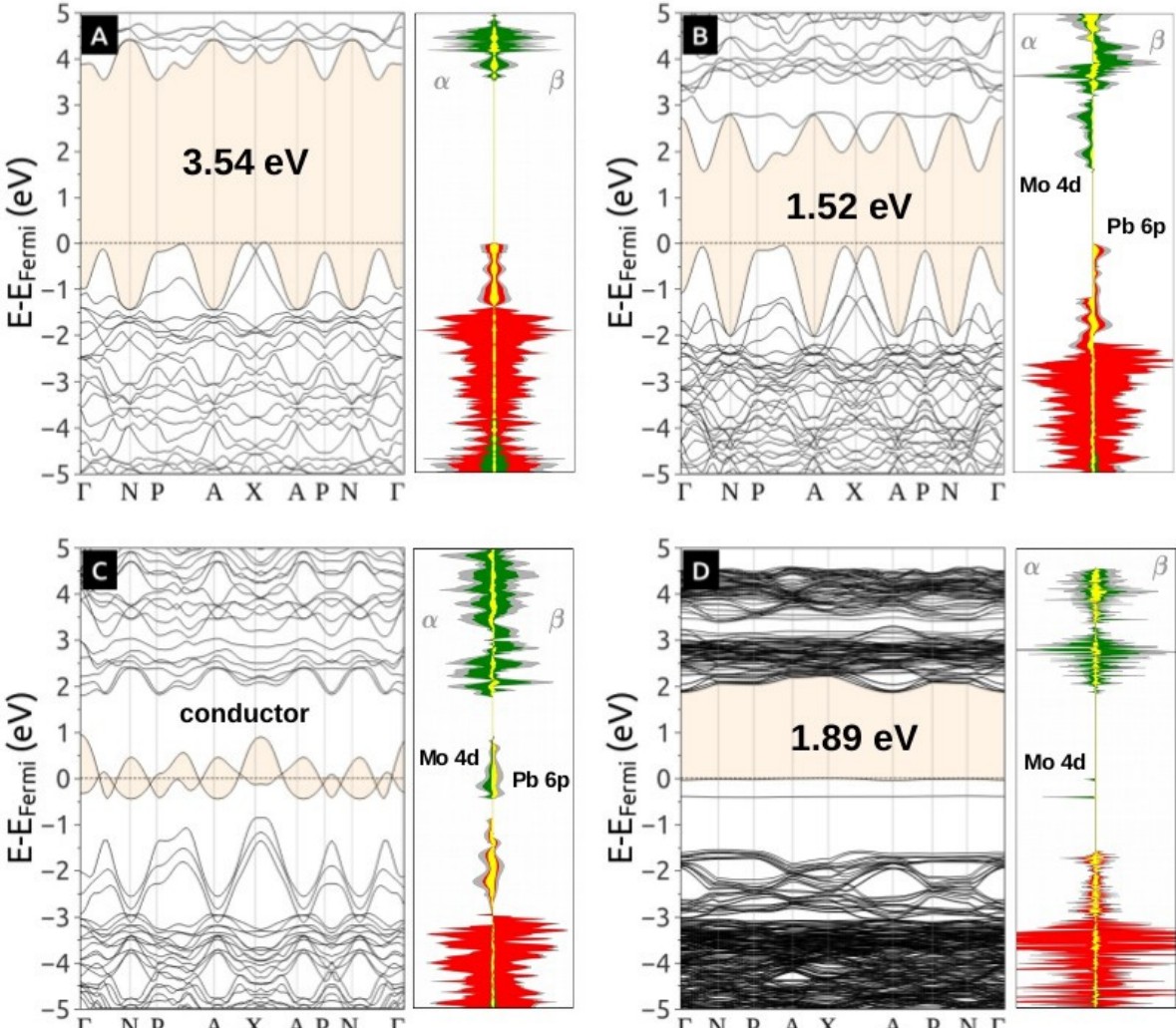

**Figure 5.** Band structure and projected density of states of stoichiometric $PbMoO_4$ in the (**A**) ground $S_0$ state, (**B**) triplet $T_1$ without symmetry broken, (**C**) triplet $T_1$ with symmetry broken (and fully relaxed), (**D**) reduced $PbMoO_{3.94}$. The horizontal red line indicates the Fermi level. In all cases, alpha and beta electrons are together in the same diagram. Projected DOS for Pb, Mo and O are colored in yellow, green and red, respectively.

Reduced $PbMoO_{3.9375}$ also presents midgap states due to trapped electrons in Mo. The main difference with stoichiometric molybdate is the oxidation degree of Mo that maintains the electrons left by the removed oxygen but preserves the crystalline order at long-range (for low $V_O$ concentration).

*3.5. FE-SEM Images of PbMoO$_4$ Nanocrystals*

Figure 6a–l shows the micrographs for samples PbMoO$_4$-1 (Figure 6a–c), PbMoO$_4$-10 (Figure 6d–f), PbMoO$_4$-30 (Figure 6g–i) and PbMoO$_4$-60 (Figure 6j–l). Several irregular polyhedral were obtained for all samples, generally composed of agglomerates, similar to those reported [6,59–61]. Moreover, these nanocrystals have been exposed to all surface planes [8,9].

Energy dispersive X-ray (EDX) analysis of PbMoO$_4$-1 (Figure 6b), PbMoO$_4$-10 (Figure 6f), PbMoO$_4$-30 (Figure 6j) and PbMoO$_4$-60 (Figure 6n) samples confirms EDX peaks of lead (Pb), molybdenum (Mo) and oxygen (O) characteristic of PbMoO$_4$. The aluminum (Al) peak identified in the EDX spectrum comes from the support used to accommodate the samples while acquiring the EDX spectrum and micrographs.

We performed size distribution by initially measuring the length of 100 PbMoO$_4$ nanocrystals using free software package ImageJ, version 1.8.0, for Windows (64-bits). The histograms plotted for the frequency and size of the measured PbMoO$_4$ nanocrystals resulted in the best adjustment for the experimental values with a log-normal function [9,14,53].

Histograms for the frequency and mean size of nanocrystals PbMoO$_4$-1, PbMoO$_4$-10, PbMoO$_4$-30 and PbMoO$_4$-60 are presented in parts of Figure 6d,h,l,p, respectively.

The increase in synthesis time shows the average particle size obtained for PbMoO$_4$-1, PbMoO$_4$-10 and PbMoO$_4$-30 as $\bar{x} = 154.90(8)$ nm, $\bar{x} = 176.73(4)$ nm and $\bar{x} = 303.02(1)$ nm, respectively. However, there was a decrease in $\bar{x} = 287.41(1)$ nm when we analyzed the nanocrystals synthesized at 60 min (PbMoO$_4$-60). Therefore, based on these results, it is suggested that a synthesis time of 60 min causes dissolution of particles previously condensed by the Ostwald ripening process.

Adjustment of the log-normal curve resulted from the nanocrystals that comprised the sample PbMoO$_4$-1, $\bar{x} = 154.90(8)$ nm. However, it is composed mainly of 25% from 117–141 nm, 30% between 146–197 nm, 18% (90–113 nm) and 11% between 201–225 nm. Furthermore, the average of the nanocrystals length $\bar{x} = 176.73(4)$ nm for the PbMoO$_4$-10 nanocrystals is mainly composed of 44% (113–165 nm), 29% (174–224 nm), 11% (234–285 nm) and 6% (54–105 nm). For PbMoO$_4$-30 and PbMoO$_4$-60 samples, the average nanocrystal lengths were $\bar{x} = 303.02(3)$ nm and 287.41(1) nm, respectively. In the first case, the percentage of nanocrystals in the interval from 0 to 150 nm, 150 to 300 and 300 to 460 are in this order: 17%, 45% and 13%, while, for sample PbMoO$_4$-60, 22% of nanocrystals exhibit size length between 0 and 200 nm, 52% from 200 to 400 nm, 18% from 400 to 600 nm and 8% have size length more than 600 nm.

*3.6. Photocatalysis of RhB and RBBR Dyes with PbMoO$_4$ Nanocrystals*

We evaluated the catalytic performance of the PbMoO$_4$ nanocrystals by heterogeneous photocatalysis using solutions of Rhodamine B—RhB (cationic) and Brilliant Blue of Remazol Blue—RBBR (anionic) dyes in the presence of nanocrystals under UV radiation ($\lambda = 253$ nm). Photocatalytic assays performed with the RhB dye (50 mL, $1 \times 10^{-1}$ mol L$^{-1}$) in the absence (photolysis), and presence of the PbMoO$_4$ nanocrystals is presented in parts of Figure 7a–f.

It is clear that photolysis, that is, the effect of UV radiation in the absence of catalysts, did not significantly affect the mechanism of degradation of RhB dye molecules (Figure 7a) over 90 min exposure. The maximum absorbance at wavelength 554 nm practically did not decrease (16.38%) after 90 min of exposure. However, in the presence of PbMoO$_4$ nanocrystals (Figure 7b–e), we can notice the decrease in the said absorption maximum, a result of the oxidative processes promoted by the excitation/recombination mechanism associated with electron/hole pair ($\ominus \leftrightarrow \oplus$), between the bands (BV and BC) in the PbMoO$_4$ nanocrystals [3,12,13], finally resulting in formation of species with high oxidizing potential that attack the molecules of RhB dye, leading to total mineralization.

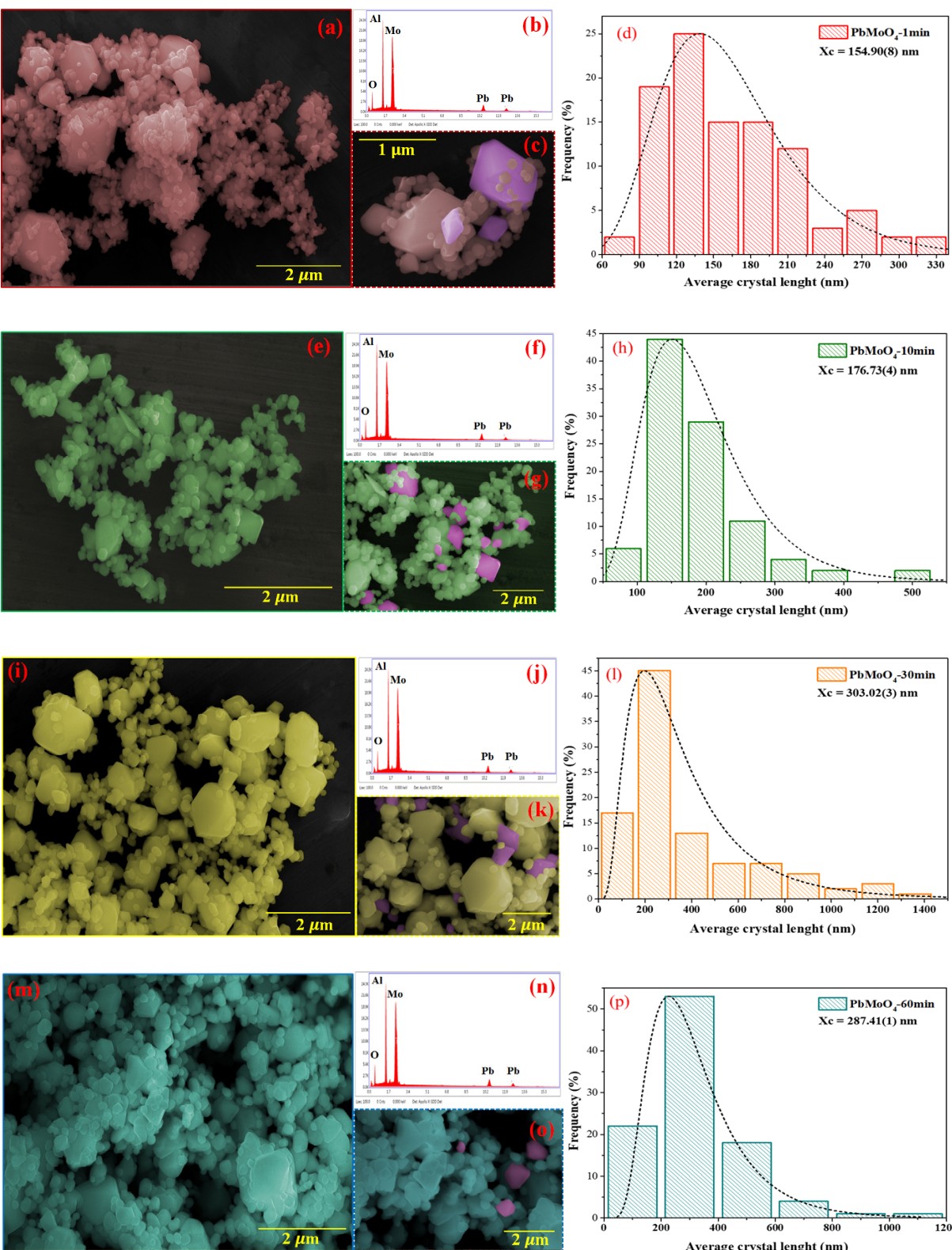

**Figure 6.** FE-SEM and EDX of PbMoO$_4$ nanocrystals synthesized at 1 min (**a–c**), 10 min (**e–g**), 30 min (**i–k**) and 60 min (**m–o**). The figures (**d,h,l,p**) correspond to the histogram of the particle size of PbMoO$_4$.

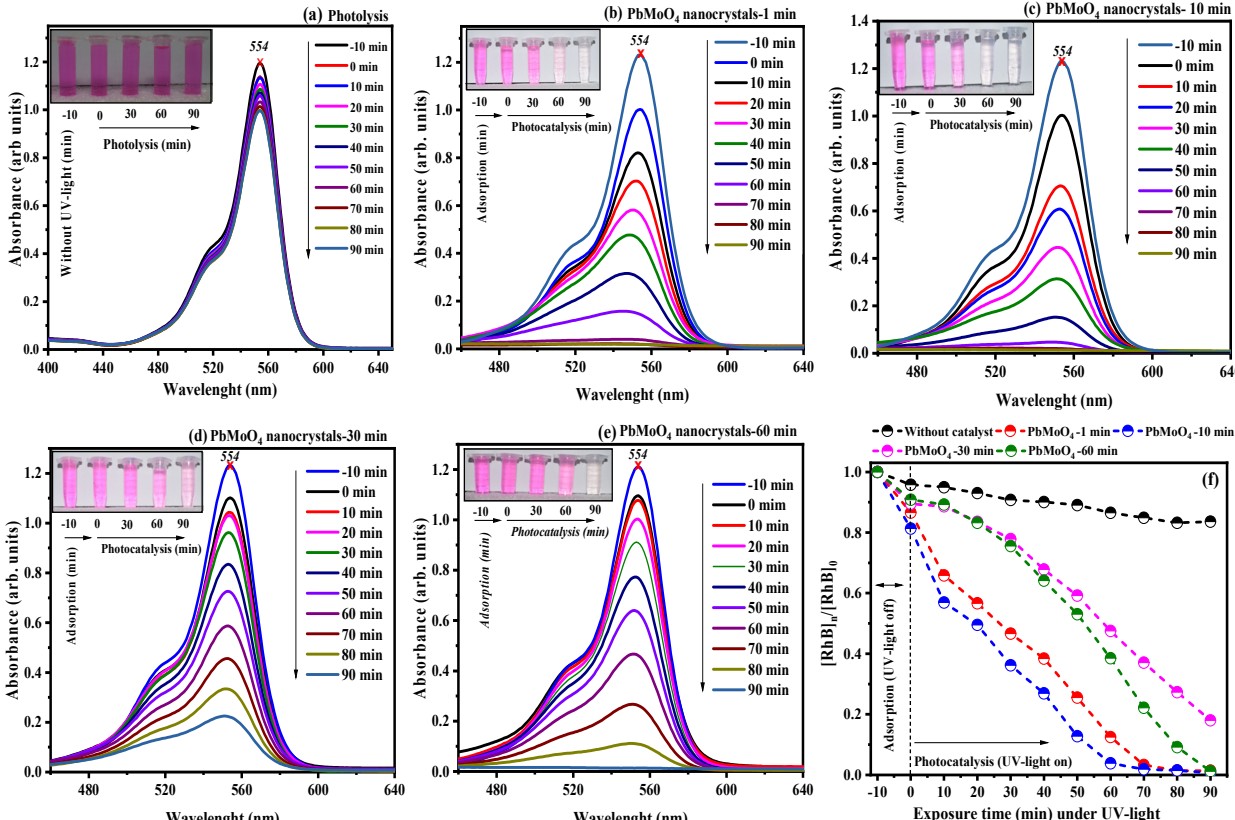

**Figure 7.** The evolution of photocatalytic degradation of RhB dye (cationic) under (**a**) UV light and (**b**–**e**) using the PbMoO$_4$ nanoparticles. Figure (**f**) represents all experiments' concentration vs. exposure time under UV light.

Figure 7f summarizes the catalytic profile evolved in the degradation of the RhB dye molecules when exposed to UV radiation (photolysis) and in the presence of the synthesized PbMoO$_4$ nanocrystals at different times of synthesis. The catalytic process evolution in the PbMoO$_4$ nanocrystal's presence followed a descending order of catalytic activity: PbMoO$_4$-10 (100%) > PbMoO$_4$-1 (100%) > PbMoO$_4$-60 (100%) > PbMoO$_4$-30 (81.99%).

There was a higher rate of degradation in the RhB dye molecules with the nanocrystals synthesized at 10 min (PbMoO$_4$-10), resulting from the high efficiency in the oxidative processes that occurred with these materials within 60 min of exposure under UV radiation. In addition, there was significant adsorption of RhB dye molecules by the nanocrystals during the initial 10 min in the absence of UV radiation, resulting from the electrostatic attraction of the dye molecules (cationic) by the presence of negative partial charges on the surface of the nanostructures [62].

In contrast, in the presence of the RBBR dye, the synthesized PbMoO$_4$ nanocrystals did not promote significant adsorption during the 10 min absence of UV radiation. Therefore, we confirm the presence of negative surface charges on the surface of the nanocrystals considering that the molecules of the RBBR dye have an anionic character when in an aqueous solution, causing electrostatic repulsion and minimum adsorption on the surface of the material [63].

The catalytic profile exhibited for the photocatalysis of the RBBR dye was similar to that observed for the RhB dye, as shown in Figure 8a–f. However, high catalytic activity occurred for the nanocrystals synthesized by the MH method in 10 min at 100 °C (PbMoO$_4$-10). Therefore, a rapid band reduction with a maximum of 592 nm at 70 min is correlated with broken bonds characteristic of the electronic transitions of the aromatic rings in the RBBR dye [3,12,13].

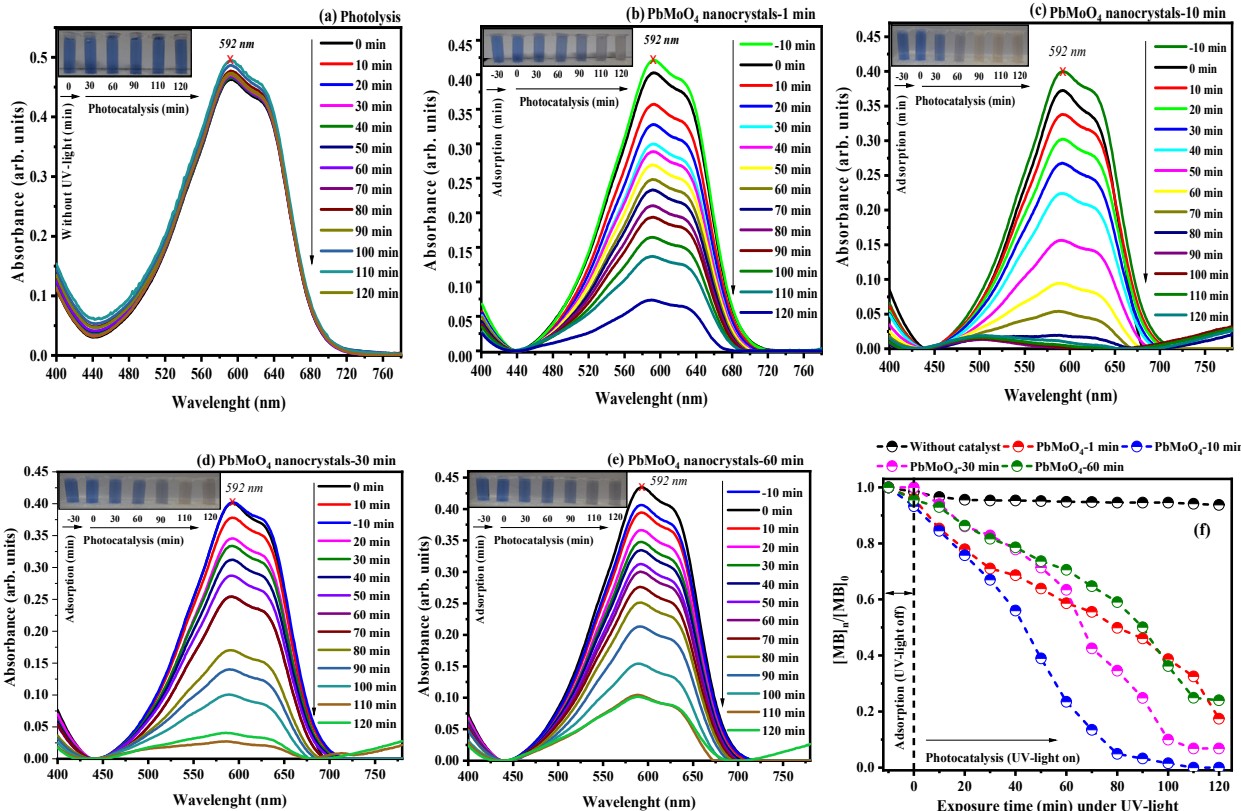

**Figure 8.** The evolution of photocatalytic degradation of RBBR dye (anionic) under (**a**) UV light and (**b**–**e**) using the PbMoO$_4$ nanoparticles. Figure (**f**) represents all experiments' concentration vs. exposure time under UV light.

The kinetic study for the catalytic processes promoted by photolysis and in the semiconductor's presence was performed by adjusting the values experimentally obtained by a pseudo-first-order function described by Langmuir–Hinshelwood, as presented in Equation (4) [56].

$$ln\left(\frac{c}{c_0}\right) = kt \tag{4}$$

$C_0$ and $c$ are the concentrations of the dyes at the initial time and throughout the catalytic process, respectively—more precisely, over 90 min for the RhB dye and 120 min for the RBBR dye. At the same time, $k$ and $t$ are the velocity constant and the time (min) of exposure to UV radiation, respectively [64].

Table 2 summarizes the results of the $k_{app}$ and half-life time ($t_{1/2}$) constants for the reactions involving the photodegradation of the RhB and RBBR dye molecules in the presence and absence of the PbMoO$_4$ nanocrystals.

Based on the summarized results from Table 2, it is confirmed that the photocatalytic performance for sample PbMoO$_4$-10 is more effective than studied samples for both dyes studied. Thus, the respective sample is 29.2 and 51.8 times more effective than photolysis for RhB and RBBR dyes. Moreover, the decreasing photocatalytic performance among the samples against the RhB dye is PbMoO$_4$-10 (26.77 × 10$^{-3}$ min$^{-1}$) > PbMoO-PbMoO$_4$-1 (26.77 × 10$^{-3}$ min) > PbMoO$_4$-30 (13.41 × 10$^{-3}$ min$^{-1}$) > PbMoO$_4$-60 (13.38 × 10$^{-3}$ min$^{-1}$). On the other hand, it can be seen that the obtained results for the photodegradation activity of studied samples over RBBR dye is PbMoO$_4$-10 (18.87 × 10$^{-3}$ min$^{-1}$) > PbMoO-PbMoO$_4$-30 (9.06 × 10$^{-3}$ min) > PbMoO$_4$-1 (8.64 × 10$^{-3}$ min$^{-1}$) > PbMoO$_4$-60 (7.72 × 10$^{-3}$ min$^{-1}$). Two main factors can be pointed out for sample PbMoO$_4$-10, which could be associated with the photocatalytic performance against the RhB and RBBR dyes. Through XRD analysis is confirmed the smallest crystallite size, where structural defects and oxygen vacancies

are present, while there is high adsorption capacity, as can be seen in Figures 7f and 8f, for sample PbMoO$_4$-10.

**Table 2.** Discoloration ratio (%), apparent kinetic rate constant ($k_{app}$) and half-life time ($t_{1/2}$) obtained from photocatalysis assay of RhB and RBBR with photolysis and PbMoO$_4$ nanocrystals.

| Sample | RhB Dye | | | RBBR Dye | | |
|---|---|---|---|---|---|---|
| | Disc. (%) | $k_{app} \times 10^{-3}$ (min$^{-1}$) | $t_{1/2}$ (min) | Disc. (%) | $k_{app} \times 10^{-3}$ (min$^{-1}$) | $t_{1/2}$ (min) |
| Photolysis | 16.38(7) | 1.88 | 368.6 | 6.27(5) | 0.364 | 1904.2 |
| PbMoO$_4$-1 | 100 | 26.77 | 25.8 | 82.56(5) | 8.64 | 80.2 |
| PbMoO$_4$-10 | 100 | 52.45 | 13.2 | 100 | 18.87 | 36.7 |
| PbMoO$_4$-30 | 81.99(5) | 13.41 | 51.6 | 93.18(4) | 9.06 | 76.5 |
| PbMoO$_4$-60 | 100 | 13.38 | 51.8 | 76.0 | 7.72 | 89.7 |

Legend: Disc. = discoloration rate.

Based on the results summarized in Table 2, it is confirmed that the photocatalytic performance for the PbMoO$_4$-10 sample is more effective than the studied samples for both studied dyes. Thus, the respective sample is 29.2 and 51.8 times more effective than photolysis for RhB and RBBR dyes. Furthermore, the decreasing photocatalytic performance between samples against the RhB dye is PbMoO$_4$-10 (26.77 × 10$^{-3}$ min$^{-1}$) > PbMoO-PbMoO$_4$-1 (26.77 × 10$^{-3}$ min$^{-1}$) > PbMoO$_4$-30 (13.41 × 10$^{-3}$ min$^{-1}$) > PbMoO$_4$-60 (13.38 × 10$^{-3}$ min$^{-1}$). On the other hand, it can be observed that the results obtained for the photodegradation activity of the studied samples on the RBBR dye are PbMoO$_4$-10 (18.87 × 10$^{-3}$ min$^{-1}$) > PbMoO$_4$-30 (9.06 × 10$^{-3}$ min$^{-1}$) > PbMoO$_4$-1 (8.64 × 10$^{-3}$ min$^{-1}$) > PbMoO$_4$-60 (7.72 × 10$^{-3}$ min$^{-1}$). Two main factors can be pointed out for the PbMoO$_4$-10 sample, which may be associated with the photocatalytic performance against RhB and RBBR dyes. Through XRD analysis, the smallest crystallite size is confirmed, where structural defects and oxygen vacancies are present, while the high physical adsorption capacity of the dyes on the surface of the nanocrystal, as can be seen in Figures 7f and 8f, for the PbMoO$_4$-10 sample is higher than all samples for both dyes.

The oxidative processes in aqueous solution involve excitation/recombination of electrons by the transfer of energy from the absorbed photons from the photocatalyst. Consequently, there will be formation of superoxide radicals ($O_2^{\bullet-}$), hydroxyl radicals (HO$^{\bullet}$) and holes ($h^+$) [3,12,13]. Thus, use of radical scavengers in photocatalytic experiments is an interesting approach to evaluate the performance of each radical in the catalytic processes involved [12].

The catalytic performance of oxidant species in the reactional process, that is, the absence and presence of the ammonium oxalate (AO) (scavenger of holes, $h^+$), parabenzoquinone—PB, superoxide radical scavenger ($O_2^{\bullet-}$) and tert-butanol alcohol—ATB (scavenger of the hydroxyl radical (HO$^{\bullet}$), was performed using the sample PbMoO$_4$-10 as photocatalyst, where the results are shown in Figure 9a (RhB) and Figure 9b (RBBR).

There was a total degradation of the RhB and RBBR dye molecules in the absence of scavenger radicals at UV radiation exposure times of 80 and 110 min, respectively. Therefore, the PBQ significantly reduces the catalytic effect when the scavengers' radicals are added, implying that the superoxide radicals significantly contribute to the catalytic events. In contrast, the hydroxyl radicals' contributions are due to the holes generated by electrons vacancy in the valence band [9,13].

We performed a redox of the PbMoO$_4$-10 nanocrystals after four catalytic cycles at 90 min (RhB) and 120 min (RBBR), measuring the catalyzed solutions' absorbance and quantifying the degradation percentage for each smoothed dye solution. Figure 10 shows the results of reusing PbMoO$_4$-10 nanocrystals during four photocatalytic cycles in the solution's presence of RhB and RBBR dyes.

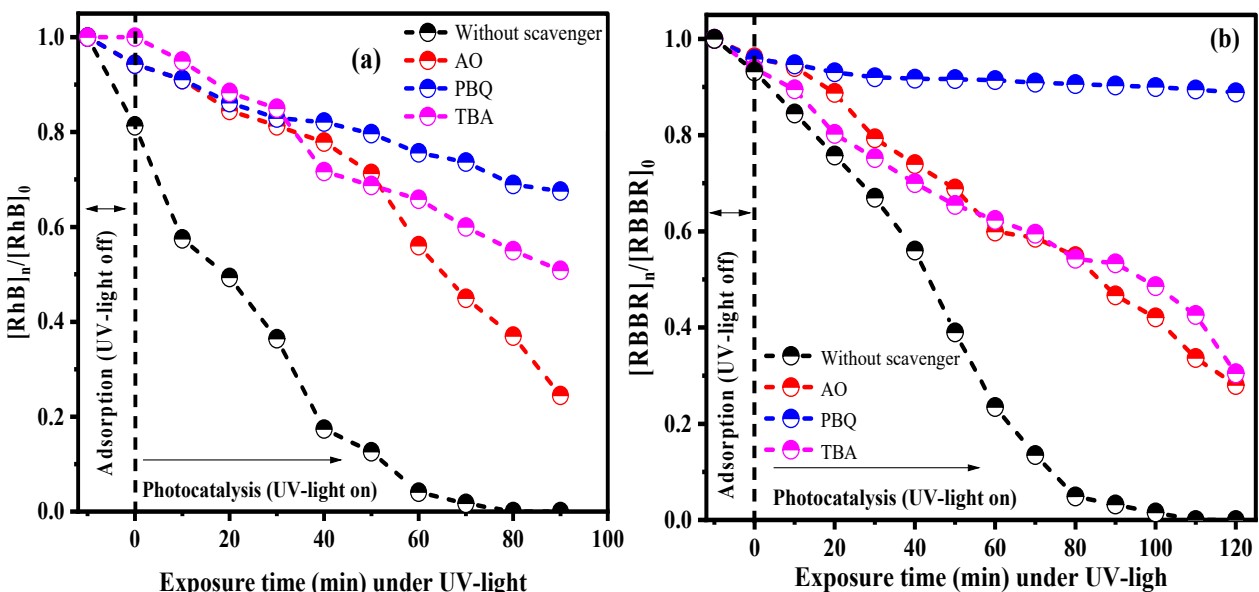

**Figure 9.** Catalytic performance of the PbMoO$_4$ nanocrystals in the photodegradation of (**a**) cationic (RhB) and an (**b**) anionic dye (RBBR) with the presence and absence of scavenger radicals (OA, PBQ and TBA).

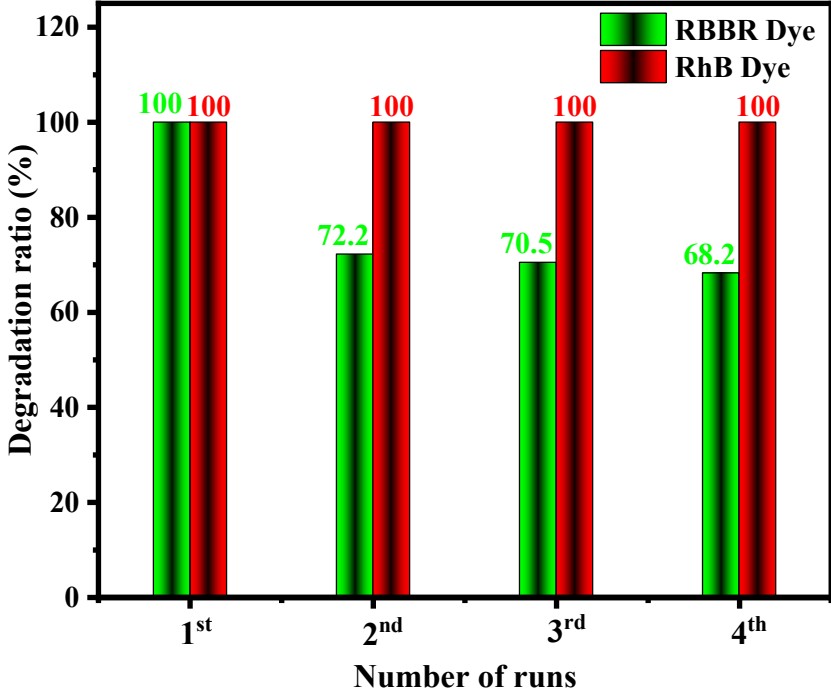

**Figure 10.** Reusability of PbMoO$_4$-10 in the photodegradation of RhB (cationic) RBBR (anionic) dye. Initial conditions = 50 mg of PbMoO$_4$-10, volume of dye = 50 mL, time = 90 min (RhB) and 120 min (RBBR).

The decolorization percentage of the RhB dye solution compared to the photocatalytic tests in the PbMoO$_4$-10 nanocrystal presence did not change significantly at the end of four photocatalytic cycles. These are close to 100% for all materials analyzed. However, there was a gradual decrease in solution discoloration percentage in the presence of RBBR dye molecules, obtaining 100, 72.107, 70.775 and 68.113% for the four cycles, referring to the first, second, third and fourth cycles, respectively.

Figure 11 shows the schematic that summarizes the proposed mechanism for the catalytic process involving PbMoO$_4$ nanocrystals synthesized at different synthesis times ($t$ = 1, 10, 30 and 60 min) in the photodegradation of RhB and RBBR dye molecules.

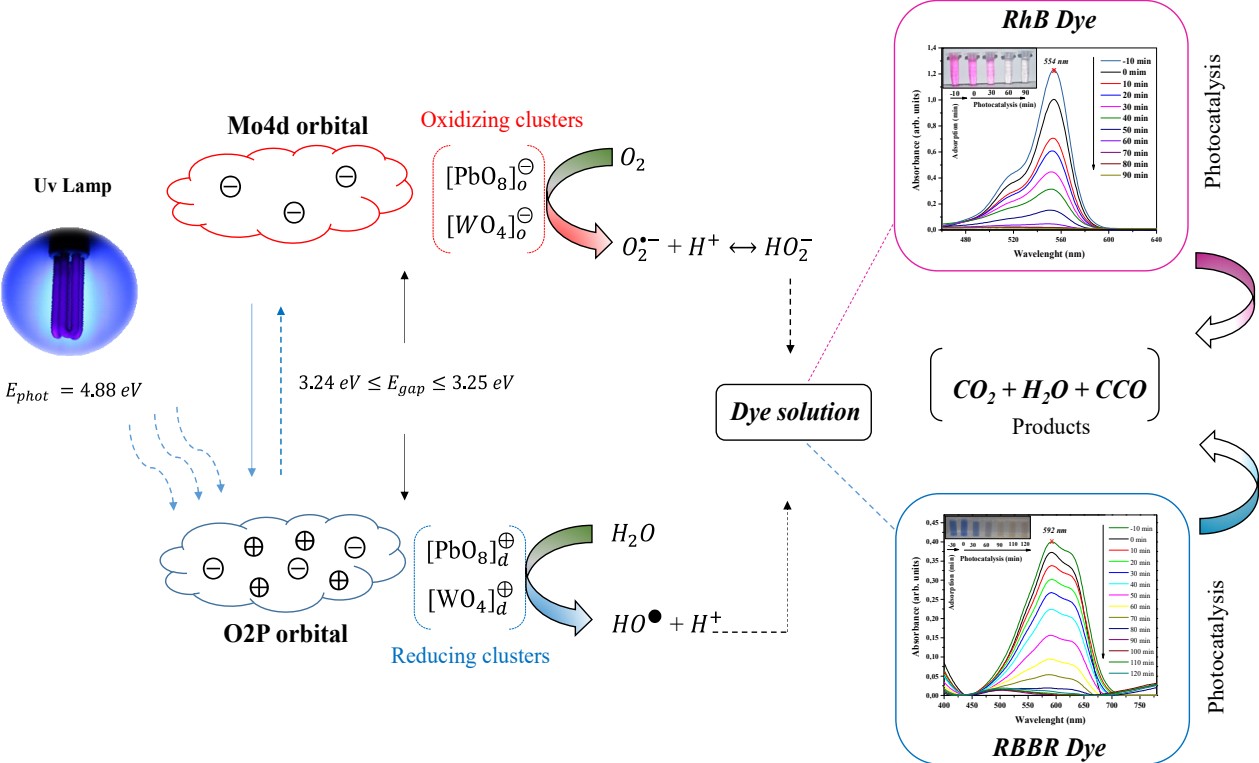

**Figure 11.** Scheme of the photocatalytic process after absorption of UV light, resulting in degradation of RhB (cationic) and RBBR dye (anionic) by oxidant effect.

The adsorption mechanism of the molecules on the surface of the nanocrystals initially promotes equilibrium by electrostatic attraction [41,42,63,65]. When irradiated with the three UV lamps ($\lambda$ = 253 nm; $E_{photon}$ = 4.88 eV), the nanocrystals absorb the photon energy with bandgap ($E_{gap}$) value ($3.24 \leq E_{gap} \leq 3.25$), resulting in excitation/recombination of the electrons ($\ominus$), thus generating the holes ($h^+ = \oplus$) [17].

The crystalline defects and deformations in the length and angle of the bonds in the deltahedral [PbO$_8$] and tetrahedral [MoO$_4$] clusters play a fundamental role in the efficiency of the electron/hole pairing process. In this way, we believed that the order/structural disorder exhibits different types of order clusters ([PbO$_8$]$_o$, [MoO$_4$]$_o$) and disordered ([PbO$_8$]$_d$, [MoO$_4$]$_d$) [56]. Other factors can be considered in this process, among which we highlight the presence of vacancies of oxygen atoms, energy surfaces ($E_{surf}$), crystallite size and microdeformations, promoting insertion of intermediate levels between the valence band and conduction band [8,17].

Based on the obtained results, mainly in the X-ray diffraction analysis and photocatalytic experiments, we proposed that: (i) the smallest crystallite size; (ii) high adsorption capacity and (iii) density of defects in the structure for sample PbMoO$_4$-10 promote polarization at the short and long range of the ordered clusters ([PbO$_6$]$_o^x$/[MoO$_4$]$_o^x$ and [PbO$_6$]$_o^x$/[MoO$_4$]$_o^x$) and the disordered ([PbO$_6$]$_d^x$/[MoO$_4$]$_d^x$ and [PbO$_6$]$_d^x$, [MoO$_4$]$_d^x$) structures of the nanocrystals [9]. Consequently, the efficient excitation/recombination of electrons in ordered ([PbO$_6$]$_o^\ominus$/[MoO$_4$]$_o^\ominus$ and [PbO$_6$]$_o^\oplus$/[MoO$_4$]$_o^\oplus$) and disordered ([PbO$_6$]$_o^\ominus$, [MoO$_4$]$_o^\ominus$ and [PbO$_6$]$_o^\oplus$, [MoO$_4$]$_o^\oplus$) clusters when they absorb photons from UVc-light source, as shown in Equations (5)–(8) [14].

$$[\text{PbMoO}_4]_{(\text{defects})} \rightarrow [\text{PbO}_8]_o^x + [\text{PbO}_8]_d^x \tag{5}$$

$$[\text{PbMoO}_4]_{(\text{defects})} \rightarrow [\text{MoO}_8]_o^x + [\text{MoO}_8]_d^x \tag{6}$$

$$\overset{h\nu\,=\,4.88\text{ eV}}{\rightarrow}\; [\text{PbO}_8]_o^x + [\text{PbO}_8]_d^x \rightarrow [\text{PbO}_8]_o^{\oplus} + [\text{PbO}_8]_d^{\ominus} \tag{7}$$

$$\overset{h\nu\,=\,4.88\text{ eV}}{\rightarrow}\; [\text{MoO}_8]_o^x + [\text{MoO}_8]_d^x \rightarrow [\text{MoO}_8]_o^{\oplus} + [\text{MoO}_8]_d^{\ominus} \tag{8}$$

The adsorption equilibrium in the absence of UV radiation conduced to adsorption of water molecules and RhB and RBBR dyes on the surface of the $\text{PbMoO}_4$ nanocrystals [42]. Thus, the oxidative processes from the pair electron/hole ($\ominus \leftrightarrow \oplus$) recombination induces formation of hydroxyl radicals ($\text{HO}^\bullet$) and $\text{H}^+$ ions through oxidation of water molecules by holes from the valence band [12,13]. The processes mentioned above are shown in Equations (9)–(12).

$$[\text{PbO}_8]_o^{\ominus} + [\text{PbO}_8]_d^{\oplus} + \text{H}_2\text{O} \rightarrow [\text{PbO}_8]_o^{\ominus} + [\text{PbO}_8]_d^{\oplus} \ldots \text{H}_2\text{O}_{(ads)} \tag{9}$$

$$[\text{PbO}_8]_o^{\ominus} + [\text{PbO}_8]_d^{\oplus} \ldots \text{H}_2\text{O}_{(ads)} \rightarrow [\text{PbO}_8]_o^{\ominus} + [\text{PbO}_8]_d^x + \text{H}^+ + \text{HO}^\bullet \tag{10}$$

$$[\text{MoO}_4]_o^{\ominus} + [\text{MoO}_4]_d^{\oplus} + \text{H}_2\text{O} \rightarrow [\text{MoO}_4]_o^{\ominus} + [\text{MoO}_4]_d^{\oplus} \ldots \text{H}_2\text{O}_{(ads)} \tag{11}$$

$$[\text{MoO}_4]_o^{\ominus} + [\text{MoO}_4]_d^{\oplus} \ldots \text{H}_2\text{O}_{(ads)} \rightarrow [\text{MoO}_4]_o^{\ominus} + [\text{MoO}_4]_d^x + \text{H}^+ + \text{HO}^\bullet \tag{12}$$

The excited electrons of the valence band are distributed over the surface of the nanocrystals before returning to the valence band. However, the electrons are captured in the presence of oxygen molecules ($\text{O}_2$) adsorbed on the $\text{PbMoO}_4$ nanocrystals' surface, resulting in formation of superoxide radicals ($\text{O}_2^{\bullet-}$) [66]. Thus, the oxidation mechanism of the water molecules through the holes leads to hydronium ions ($\text{H}^+$) formation that, in the presence of superoxide radicals, gives rise to hydroperoxide radicals ($\text{HO}_2^-$) [62]. The reactions involved are presented in Equations (13)–(18).

$$[\text{PbO}_8]_o^{\oplus} + [\text{PbO}_8]_d^{\ominus} + \text{O}_{2(ads)} \rightarrow [\text{PbO}_8]_o^{\oplus} + [\text{PbO}_8]_d^{\ominus} \ldots \text{O}_{2(ads)} \tag{13}$$

$$[\text{PbO}_8]_o^{\oplus} + [\text{PbO}_8]_d^{\oplus} \ldots \text{O}_{2(ads)} \rightarrow [PbO_8]_o^{\oplus} + [PbO_8]_d^x + \text{O}_2^{\bullet-} \tag{14}$$

$$\text{O}_2^{\bullet-} + \text{H}^+ \rightarrow \text{HO}_2^- \tag{15}$$

$$[\text{MoO}_4]_o^{\oplus} + [\text{MoO}_4]_d^{\ominus} + \text{O}_{2(ads)} \rightarrow [\text{MoO}_4]_o^{\oplus} + [\text{MoO}_4]_d^{\ominus} \ldots \text{O}_{2(ads)} \tag{16}$$

$$[\text{MoO}_4]_o^{\oplus} + [\text{MoO}_4]_d^{\ominus} \ldots \text{O}_{2(ads)} \rightarrow [\text{MoO}_4]_o^{\oplus} + [\text{MoO}_4]_d^x \ldots \text{O}_{2(ads)} \tag{17}$$

$$\text{O}_2^{\bullet-} + \text{H}^+ \rightarrow \text{HO}_2^- \tag{18}$$

The oxidizing species ($\text{HO}^\bullet$ and $\text{HO}_2^-$) generated in the described processes have a high oxidizing potential in the reaction medium. In the presence of RhB and RBBR dye solutions, they promote oxidation of organic molecules [3,11,64]. Their degradation by oxidizing species (Equations (19)–(20)) results in total mineralization, that is, formation of low-molecular-weight non-toxic compounds ($\text{H}_2\text{O}$ e $\text{CO}_2$) and colorless species (CCO).

$$\overset{h\nu\,=\,4.88\text{ eV}}{\rightarrow}\; \text{RhB} + \text{HO}_2^- + \text{HO}^\bullet \rightarrow \text{H}_2\text{O} + \text{CO}_2 + \text{CCO} \tag{19}$$

$$\overset{h\nu\,=\,4.88\text{ eV}}{\rightarrow}\; \text{RBBR} + \text{HO}_2^- \text{HO}^\bullet \rightarrow \text{H}_2\text{O} + \text{CO}_2 + \text{CCO} \tag{20}$$

The adsorbed molecules on the $PbMoO_4$ nanocrystals' surface during adsorption equilibrium are also oxidized directly by the action of the holes present in the valence band [3,9,13,17,58], as described in Equations (21)–(26).

$$[PbO_8]_o^{\ominus} + [PbO_8]_d^{\oplus} + RhB_{(ads)} \rightarrow [PbO_8]_d^{\oplus} + [PbO_8]_o^{\ominus} \ldots RhB_{(ads)} \tag{21}$$

$$[PbO_8]_o^{\ominus} + [PbO_8]_d^{\oplus} + RhB_{(ads)} \rightarrow [PbO_8]_o^{\ominus} + [PbO_8]_d^{x} + RhB^{+-} \tag{22}$$

$$RhB^{+-} \overset{h\nu=4.88\ eV}{\Rightarrow} H_2O + CO_2 + CCO \tag{23}$$

$$[MoO_4]_o^{\ominus} + [MoO_4]_d^{\oplus} + RBBR_{(ads)} \rightarrow [MoO_4]_o^{\ominus} + [MoO_4]_d^{\oplus} \ldots RBBR_{(ads)} \tag{24}$$

$$[MoO_4]_o^{\ominus} + [MoO_4]_d^{\oplus} \ldots RBBR_{(ads)} \rightarrow [MoO_4]_o^{\ominus} + [MoO_4]_d^{x} + RBBR^{+-} \tag{25}$$

$$RBBR^{+-} \overset{h\nu=4.88\ eV}{\Rightarrow} H_2O + CO_2 + CCO \tag{26}$$

where CCO is the colorless by product of low molecular weight.

## 4. Conclusions

We synthesized $PbMoO_4$ nanocrystals with a high degree of crystallinity by microwave hydrothermal method at synthesis times of 1, 10, 30 and 60 min at 100 °C. The theoretical study by DFT resulted in excellent correlation between experimental and theoretical values, agreeing with the information presented in Raman vibrational spectroscopy and UV–vis spectroscopy by diffuse reflectance with DOS and the proposed catalytic process. The nanoparticles only exhibited tetragonal phases with particle size distributions between 154 and 303 nm, presenting, in general, irregular polyhedrons with 10 well-defined faces. All the $PbMoO_4$ nanoparticles synthesized presented catalytic efficiency in decolorizing the Rhodamine B and Brilliant Blue of Remazol Blue (RBBR) dyes; however, the best catalytic results obtained for the nanoparticles synthesized were at 10 min. In the oxidative processes, we verified that the holes generated by absence of electrons and the superoxide radicals were the main species involved in the degradation mechanism of the dye molecules when the $PbMoO_4$ nanocrystals were exposed to UV radiation in an aqueous solution.

**Supplementary Materials:** The following supporting information can be downloaded at: https://www.mdpi.com/article/10.3390/colorants2010008/s1, Figure S1: UV–vis spectra of $PbMoO_4$ nanocrystals synthesized at (a) 1 min, (b) 10 min, (c) 30 min and (d) 60 min by MH under 100 °C. Figure S2: Calculated electronic (including dispersion) energy for stoichiometric $PbMoO_4$ in singlet or triplet spin-polarized solutions. The up vertical arrow indicates spin polarization without atomic positions and lattice parameters relaxation. Despite the low energy difference, the energy may increase or decrease if the electronic correlation raises internal pressure. At the same time, the lattice is frozen, and the cations have partial reduction (delocalized solution). The energy decreases substantially with internal symmetry broken (vertical transition, blue curve) and a little more with lattice relaxation; Figure S3: Electronic spin density isosurfaces of (a) stoichiometric $PbMoO_4$ in the spin-polarized triplet solution and (b) the supercell of reduced $PbMoO_{3.94}$, where only the local symmetry is broken with the oxygen vacancy ($V_0$) formation. Figure (a) shows that the conventional unit cell was replicated in the [100] plan; Table S1: Rietveld refinement results of $PbMoO_4$ nanocrystals; Table S2: Experimental and theoretical active Raman modes of $PbMoO_4$ nanocrystals synthesized and reported by the literature; Table S3: Lattice parameters (Å and Å$^3$), total energy differences per unit cell (eV) and electronic band gap (eV) of stoichiometric and reduced $PbMoO_4$.

**Author Contributions:** Idea and writing: F.N., M.d.N. and J.T.; Structural characterization by XRD analysis: O.M., G.S. and Y.L.R.; Raman and diffuse reflectance spectroscopies: P.C. and W.B.; DFT study: A.A. and J.S.; Photocatalytic study: F.N., J.M.D.M. and Y.L.R. All authors have read and agreed to the published version of the manuscript.

**Funding:** This research was funded by Samsung Eletrônica da Amazônia Ltda., using resources from the Federal Law nᵒ 8.387/1991, being its dissemination and publicity, in accordance with following the provisions of article 39 of Decree No. 10.521/2020 for its dissemination and publicity, which the APC for development of this project was R$ 8.6 M (reais).

**Institutional Review Board Statement:** Not applicable.

**Informed Consent Statement:** Not applicable.

**Data Availability Statement:** Not applicable.

**Acknowledgments:** The authors would like to thank the Coordenação de Aperfeiçoamento de Pessoal de Nível Superior—CAPES for financial support, as well as Central Analítica, do IFAM, Campus Manaus Centro and Fundação de Amparo à Pesquisa do Estado do Amazonas—FAPEAM. This article is the result of the RD&I Lively IoT Catheter project carried out by the University of the State of Amazonas (UEA) in partnership with Samsung Eletrônica da Amazônia Ltda. using resources from Federal Law nᵒ 8.387/1991 following the provisions of article 39 of Decree No. 10.521/2020 for its dissemination and publicity.

**Conflicts of Interest:** The authors declare no conflict of interest.

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
