# Peer review of "Photocatalytic Properties of PbMoO4 Nanocrystals against Cationic and Anionic Dyes in Several Experimental Conditions"

_2079-6447, doi:10.3390/colorants2010008_

Round 1

Reviewer 1 Report

The authors describe MW synthesis of PbMoO4 nanocrystals and its use for dye degradation.

In my opinion, the manuscript is very poorly written and difficult to follow. It is also of my opinion that the authors presented well the computational aspects of research but severely lacked the experimental part and especially the photocatalytic aspect.

The major critiques are:

-       Bad English language, which is not suitable for an international journal,

-       Bad photocatalytic experimental design,

-       Bad description of the experimental results,

-       No attainable conclusions presented.

My suggestion is that authors elaborate more on the computational part, severely improve the English language, and try to send the manuscript to a more appropriate journal.

Specific comments are as follows.

1.    Several corrections are advised for abstract. No need to report techniques used, unless they are the core of the research, or they are special. The whole abstract doesn't inspire further reading. The info given here is average, that is, something you would report in the results.

2.    I would skip the whole first paragraph of introduction (starting line 48). It is too theoretical and doesn't inspire. Most of the needed info is given in the next paragraph anyway.

3.    In section 2.1 what was the MW power. This is important due to the heating effect of high vs low power.

4.    The concentration of Remazol Brilliant Blue R dye was 200 mg/L. This is a lot and is 40 times larger than concentration of RhB. Although its extinction coefficient is 5000–10000 M-1cm-1, the coefficient of RhB can go to 75000–106000 M-1cm-1, this is only important for instrumentation. In terms of what the catalyst surface gets, the difference between 4.7 and 200 ppm is too large to be comparable.

5.    Figure 1: It would be better to plot in reverse order. That is, 1 min the lowest and 60 min the highest. I assume you want to show growth kinetics. It is not visible from this plot. The series should be vertically translated more in order to better compare the height / width of the peaks.

6.    Line 292: This should be written in experimental part! What PbMoO4-10 means is known to reader by now. But it should be clearly stated in experimental part before!

7.    Line 300: The authors state: “It can be associated with the heated process and pressure attributed in the microwave-hydrothermal system by different synthesis times.” What was the pressure then? Did you measure it? How was the temperature measured? IR? The instrument for MW synthesis is not described!

8.    Figure 4: Simmertic. Should be symmetric.

9.    First paragraph of section 3.5 is redundant (line 447).

10.  Paragraph in line 481 is too detailed. No need to explain the technique. Just write log-normal.

11.  Line 499: The authors say: “there was a decrease for ?Ì… = 287.41(1) nm when the nanocrystals synthesized at 60 min.” What is the reason for this. You stated previously that coalescence of multiple particles is responsible for increase of size after 10 min. So, what happens from 30 to 60 min to observe a decrease?

12.  Figure 7: Taking 10 consecutive 5 mL aliquots means you have no more solution left. This drastically changes the reaction conditions! Such results are not reliable. Same goes for figure 8.

13.  The paragraph starting at line 578 only restates what is written in Table 2. This is redundant and distracting to the reader! There is no description on the trends. It seems at least partially connected to the size of nanocrystals, yet nothing is said abot that!

14.  Which catalyst was used in figure 9?

15.  Why don't you elaborate the idea starting at line 641 more? Did you try to find any correlations between the mentioned properties and photocatalytic activities?

Author Response

Dear Reviewer #1,

In the attached file are the point-by-point response to the questions/comments raised in the analysis of the manuscript. We would like to thank you for your notable contribution to our work, which is presented as a decisive point for a better understanding of our study. Thanks.

Reviewer 2 Report

In the manuscript entitled "Photocatalytic properties of PbMoO4 nanocrystals against cationic and anionic dyes at several experimental conditions" submitted by Francisco Nobre et al., the authors described the preparation and testing of PbMoO4 nanocrystals produced by microwave hydrothermal method. In the work, the structure of nanocrystallites was thoroughly characterized, optical properties and surface morphology were presented. The photocatalytic activity of the material against cationic (rhodamine B) and anionic (Remazol Brilliant Blue R) dye under the influence of UV light was tested. The work is experimental and calculational in nature.

The manuscript is clear and generally well-written, even if some improvements could be produced. For this reason, I recommend the publication of this manuscript on Colorants after the following minor revisions:

- In the Introduction section, (line 88 and 92), cited references are in a different format than the others. Please, unify the citations;

- In the Introduction section I propose add some newest references (2020 – 2022 years);

- Please check the numbering of Tables and Figures as there are errors in the text;

- In Table 2 the column header is "103" it should be "10-3";

- The EDS spectra in Fig. 6 are illegible. This should be improved;

- In the paragraph (p. 14, lines 501-506) the histograms of the particle size distribution of only two samples are interpreted (PbMoO4-1 i PbMoO4-10). Please complete this description with the others, or write why these two are described.

Author Response

Dear Reviewer #2,

In the attached file are the point-by-point response to the questions/comments raised in the analysis of the manuscript. We would like to thank you for your notable contribution to our work, which is presented as a decisive point for a better understanding of our study. Thanks.

Reviewer 3 Report

In the study Photocatalytic properties of PbMoO4 nanocrystals against cationic and anionic dyes at several experimental conditions, the authors describe the synthesis of PbMoO4 nanocrystals by microwave hydrothermal method at different times (1, 10, 30 and 60 min). The obtained materials were subjected to various characterization techniques such as XRD, SEM, Raman and UV-Visible spectroscopy. Finally, the photocatalytic performance of the synthesized samples was evaluated in degradation of different dyes (Rhodamine B and Remazol Brilliant Blue R) under UV light irradiation. The article is well-edited and interesting. Therefore, it is suitable for publication in the journal “Colorants” after addressing the following comments:

·        Table 1, row 1 – „MH” should be „MW”.

·        Figures 7 (a-e) and Figures 8 (a-e) may be helpful for some readers, but they are not key figures of your study. So, I recommend you these figures to be added separately, in an additional file, like Supplementary data. I think that Figure 7 (f) and Figure 8 (f) are suggestive and sufficient to show the photocatalytic performances of the studied materials.

·        What is the novelty of this study? Please, emphasize it in the conclusion part.

Author Response

Dear Reviewer #3,

In the attached file are the point-by-point response to the questions/comments raised in the analysis of the manuscript. We would like to thank you for your notable contribution to our work, which is presented as a decisive point for a better understanding of our study. Thanks.

Round 2

Reviewer 1 Report

Several questions have been answered in the revised version. However, there are still some problematic areas. 

 -  the difference between 4.7 and 200 ppm between the dyes is too large to be comparable

 - in Fig.1 the increase in peaks is still difficult to see. Please shrink the amplitudes more and translate vertically even more.

 - please check the spelling once more. Line 118 says "X-ray diffraction (DRX)" which is clearly a mistake.

 I sincerely hope that the sample taken was 0.5 mL and not 5 mL. For this a µ-cuvette is needed with a volume ~0.7 mL.

Author Response

Dear reviewer, the point-by-point response to the all questions is available in the .doc attached file. Thanks.
